# Association of D-Dimer, C-Reactive Protein, and Ferritin with COVID-19 Severity in Pregnant Women: Important Findings of a Cross-Sectional Study in Northern Brazil

**DOI:** 10.3390/ijerph20146415

**Published:** 2023-07-20

**Authors:** Jenephy Thalita Rosa Paixão, Carolinne de Jesus Santos e Santos, Ana Paula Figueiredo de Montalvão França, Sandra Souza Lima, Rogério Valois Laurentino, Ricardo Roberto de Souza Fonseca, Antonio Carlos Rosário Vallinoto, Aldemir Branco Oliveira-Filho, Luiz Fernando Almeida Machado

**Affiliations:** 1Biology of Infectious and Parasitic Agents Post-Graduate Program, Federal University of Pará, Belém 66075-110, PA, Brazil; jenephypaixao@live.com (J.T.R.P.); ennilorac.carol@gmail.com (C.d.J.S.e.S.); anapaula@alexandrohup.com (A.P.F.d.M.F.); ricardofonseca285@gmail.com (R.R.d.S.F.); 2Virology Laboratory, Institute of Biological Sciences, Federal University of Pará, Belém 66075-110, PA, Brazil; saraujo@ufpa.br (S.S.L.); valois@ufpa.br (R.V.L.); vallinoto@ufpa.br (A.C.R.V.); 3Study and Research Group on Vulnerable Populations, Institute for Coastal Studies, Federal University of Pará, Bragança 68600-000, PA, Brazil; olivfilho@ufpa.br

**Keywords:** SARS-CoV-2, COVID-19, D-dimer, ferritin, C-reactive protein

## Abstract

Background: The COVID-19 pandemic has had a great impact on pregnant women due to the broad clinical spectrum of the disease. The present study investigated the profile of three biomarkers during hospital admission of pregnant women—D-dimer, C-reactive protein (CRP), and ferritin—and their correlation with the severity and outcome of COVID-19. Methods: The cross-sectional study included 226 pregnant women hospitalized in the city of Belém, Pará, Northern Brazil, from April 2020 to July 2021. Epidemiological and laboratory data were obtained from medical records, and all pregnant women underwent RT-PCR molecular testing for the detection of SARS-CoV-2. Results: In total, 121 (53.5%) were positive and 105 (46.5%) were negative for SARS-CoV-2 using RT-PCR. Most pregnant women (49.5%) with COVID-19 were between 26 and 34 years old, were residing in the interior of the state of Pará (51.2%), and were in the third gestational trimester (71.9%). In addition, 71.1% of them were admitted to the ward and 28.9% were admitted to the intensive care unit (ICU), with 90.9% surviving COVID-19. The concentrations of D-dimer (*p* = 0.0122) and ferritin (*p* ≤ 0.0001) were significantly higher among pregnant women with COVID-19, especially among those hospitalized in the ICU. Conclusion: Ferritin and D-dimer seem to serve as important biomarkers for the prognosis of COVID-19 in pregnant women, which was not observed for CRP.

## 1. Introduction

The impact of the coronavirus disease 2019 (COVID-19) pandemic on pregnant women was high in several countries. According to the Pan American Health Organization (PAHO), by March 2022, more than 365,000 pregnant women had been diagnosed with COVID-19 in the Americas, and more than 3000 of them had died [1]. In Brazil, around 22,000 pregnant women were diagnosed with COVID-19 by May 2022, of which 2026 died [2]. According to the Brazilian Observatory of COVID-19 (OOBr), race (brown or white), age (20–34 years), gestation period (third gestational trimester), and area of residence (zone urban) are important epidemiological determinants [1].

Until November 2022, the southeast and northeast regions of Brazil had the highest numbers of recorded cases of COVID-19 in pregnant women [1]. In northern Brazil, the states of Amazonas, Pará, and Rondônia had the highest numbers of recorded cases of COVID-19 in pregnant women, but the highest mortality rate was identified in Roraima, corresponding to 48% of all deaths. In the state of Pará, 869 cases of COVID-19 were detected in pregnant women, of whom 87 died. The municipalities in Pará with high numbers of cases of COVID-19 in pregnant women were Belém (*n* = 127), Ananindeua (*n* = 43), Marabá (*n* = 53), and Santarém (*n* = 40) [1].

In the guidelines for the clinical management of pregnant women with COVID-19, the Ministry of Health of Brazil (MS) [3] recommended that pregnant women with moderate or severe disease should be monitored via laboratory and imaging tests [4]. Among the recommended laboratory tests are D-dimer, C-reactive protein (CRP), and ferritin, which should be checked at hospital admission and repeated if necessary. Also, according to MS, the increase in CRP by 5 times above the upper limit of normality (ULN) and in D-dimer by 3.5 times above the ULN should be considered important laboratory findings.

It should be considered that in normal pregnancy, CRP is found at low concentrations, rising only when there is an acute complication, such as infections, inflammatory processes, or tissue damage [5]. In COVID-19, the increase in serum CRP in pregnant women has been widely reported, being suggested as an effective method of hospital screening [6]. Another serological marker of inflammation widely used in the monitoring of COVID-19 is ferritin. Ferritin elevation in pregnant women with COVID-19 is a frequent laboratory finding, especially in severe cases [7,8,9]. Regarding the analysis of coagulopathies in COVID-19, one of the main markers used is the D-dimer. Coagulation disorders in COVID-19 are characteristic of severe cases [10,11].

In healthy pregnant women, an increase in thrombin levels and a prothrombotic state are frequent findings, and an increase in this condition is common in the context of an infection; consequently, the risk of developing thrombotic complications in pregnant women with COVID-19 is high [12]. Based on this, this study described the profile of laboratory biomarkers of inflammation (CRP and ferritin) and coagulation (D-dimer) associated with COVID-19 in pregnant women hospitalized in a reference maternity hospital in the city of Belém, Pará, northern Brazil.

## 2. Materials and Methods

### 2.1. Study Population and Ethics Aspects

This is a descriptive, cross-sectional, observational, and unicentric study carried out from April 2020 to July 2021, based on clinical symptoms, laboratory, and epidemiological data of pregnant women hospitalized at the Santa Casa de Misericórdia do Pará Foundation (FSCMP) located in the city of Belém, state of Pará, northern region of Brazil.

The investigated population consisted of pregnant women hospitalized at the FSCMP who underwent testing for the detection of SARS-CoV-2. The present study was approved by the Research Ethics Committee of the FSCMP in compliance with resolutions n° 196/96 and 347/05 of the National Health Council, which provide guidelines for research involving human beings (protocol n° 4.428.886).

The FSCMP hospital is a reference hospital in the northern region of Brazil for pregnant women, puerperal women, and newborns with highly complex cases, and during the pandemic, it was a pregnant women COVID-19 reference hospital, especially for intense care treatment. According to epidemiological data from the FSCMP, from March to November 2020, 1067 cases were related to COVID-19, suspected or confirmed, involving both men and women, of which 277 were pregnant women. In 2021, 1952 suspected cases of SARS-CoV-2 infection were treated, of which 688 were confirmed, of which 149 were specifically pregnant women.

### 2.2. Study Design and Data Collection

Pregnant women hospitalized at the FSCMP who underwent RT-PCR molecular testing for the detection of SARS-CoV-2 and laboratory testing for at least one of the three investigated biomarkers were elected to participate in the study. Pregnant women were divided into four groups: (a) pregnant women with COVID-19 in ward, composed of pregnant women with confirmed SARS-CoV-2 infection via the RT-PCR method; (b) pregnant women without COVID-19 in ward, composed of pregnant women without SARS-CoV-2 infection confirmed via RT-PCR method; (c) pregnant women with COVID-19 in ICU, composed of pregnant women with confirmed SARS-CoV-2 infection via the RT-PCR method; and (d) pregnant women without COVID-19 in ICU, composed of pregnant women without SARS-CoV-2 infection confirmed via RT-PCR method. Pregnant women who did not have all the information necessary for the study in their medical records, pregnant women with neurological and/or cognitive impairment, and pregnant women who were not admitted to the FSCMP were excluded from the study.

Epidemiological, clinical, and laboratory data were accessed from electronic medical records belonging to the hospital archive. The information was recorded on physical forms and later inserted into a virtual spreadsheet, from which the statistical analysis was performed. All collections of biological material and laboratory analysis were carried out by professionals from the institution, following a standardized routine of the FSCMP.

For the diagnosis of SARS-CoV-2 infection, the RT-PCR technique was performed using a nasopharyngeal swab sample, following the instructions from the manufacturer of the Standard Q COVID-19 Ag kit (SD Biosensor, Suwon-si, Republic of Korea). Samples were collected during hospital admission. Testing was requested for women with suspected infection based on the presentation of some of the following symptoms: fever, cough, dyspnea, asthenia, myalgia, anosmia, ageusia, runny nose, headache, diarrhea, nausea, and vomiting.

The tests were performed during hospital admission by the physician in charge, according to the needs of each patient. Blood collection was performed by peripheral venipuncture, in which 10 mL of blood was collected for analysis. D-dimer was measured using the Enzyme Linked Fluorescent Assay (ELFA) technique, while CRP was measured via turbidimetry. Ferritin was measured using the electro-chemiluminescence technique, and all techniques were performed in accordance with the manufacturers’ instructions.

### 2.3. Statistical Analysis

Categorical variables were presented as frequencies (percentage), and numeric variables were presented as mean and standard deviation (SD) or median and interquartile deviation (IQR). To assess the epidemiological characteristics and levels of biomarkers, the chi-square test (χ^2^), Fisher’s exact test, or the G test were used, according to the frequencies found. The significance level adopted was 5% (*p* < 0.05). Numerical variables were evaluated in relation to normality and homogeneity of variances with the Kolmogorov–Smirnov and Levene tests, respectively. Because continuous variables do not have normal distribution and equal variances, the Kruskal–Wallis test was used, with a significance level of 5%. To measure the degree of correlation between two variables, Pearson’s correlation test was used, with a significance level of 5%. Statistical analyses were performed using Bioestat 5.3 (https://www.mamiraua.org.br/downloads/programas/ accessed on 28 June 2023) and GraphPad Prism 8.0.1 (www.graphpad.com accessed on 28 June 2023).

## 3. Results

In total, 226 pregnant women were hospitalized at FSCMP, among which 53.5% (121) were COVID-19-positive. Regarding gestational comorbidities, patients with preeclampsia were more frequent in the group without COVID-19 than in the group with COVID-19 (8.2% vs. 18.1%, *p* = 0.0450). The other epidemiological characteristics such as origin, gestational age, comorbidities, place of hospitalization, and maternal outcome did not present significant differences between the groups.

In addition, many pregnant women without COVID-19 were in the third trimester of pregnancy (74.2%, *n* = 78) and some had some pre-existing disease (14.2%, *n* = 15), such as infectious diseases (6.6%, *n* = 7) and asthma (5.7%, *n* = 6). Gestational complications were identified in 29.5% of pregnant women without COVID-19 (*n* = 31), with a higher incidence of preeclampsia (18.1%, *n* = 19) and SHSP (14.2%, *n* = 15) (Table 1).

Regarding the profile of biomarkers, there was a significant difference between D-dimer (*p* = 0.0294) and ferritin (*p* < 0.0001) in relation to the groups (with and without COVID-19) (Table 2). However, the difference observed between groups in CRP was not statistically significant (*p* = 0.3752) (Figure 1). When we also divided these groups by place of hospitalization, we found that for the D-dimer only, ICU patients with COVID-19 had significantly higher values than ICU patients without COVID-19 (1521; 2942 vs. 755.5; 864, *p* = 0.0084) (Figure 2); the other comparisons were not significant. Until our study, there has not been any data confirming the COVID-19 biomarker levels among pregnant women, in both the ward or ICU or even a comparation between these two hospital areas. In our results, we demonstrated that SARS-CoV-2-infected pregnant women in the ICU may have a greater predisposition to higher D-dimer, CRP, and ferritin levels, which are associated with severe acute respiratory syndrome symptoms; therefore, women in the SARS-CoV-2-infected group often needed to be admitted to the ICU. Mainly due to the physiological changes that occur during pregnancy, pregnant women might be more vulnerable to SARS-CoV-2 infection, thus more likely presenting severe respiratory complications, which leads to intense inflammatory response exemplified by higher D-dimer, CRP, and ferritin levels that may lead to ICU admission.

Regarding the D-dimer profile, only ICU patients with COVID-19 had significantly higher values than ICU patients without COVID-19 (1521; 2942 vs. 755.5; 864, *p* = 0.0084). Patients in the general ward did not show a significant difference (Figure 1).

Regarding the ferritin profile, patients in the ICU with COVID-19 had significantly higher values than those without COVID-19 (362.9; 352.45 vs. 99.1; 136.6, *p* < 0.0001). Among patients in the ward, there was also a significant difference, with patients with COVID-19 having higher values than those without COVID-19 (140; 235.6 vs. 79; 78.3, *p* = 0.0052). It is noteworthy that among patients with COVID-19, there was a significant difference between those in the ward and those in the ICU (140; 235.6 vs. 362.9; 352.4, *p* = 0.0196), but the same did not occur in patients without COVID-19, where there was no difference between those in the ward and those in the ICU.

The four groups were also analyzed regarding serum levels of laboratory biomarkers, which were classified as normal (within the reference value), altered (increase of up to 2× the reference value for D-dimer and ferritin and up to 2× for CRP), and very altered (above 2× the reference value for D-dimer and ferritin and above 2× for CRP) (Table 3).

Regarding D-dimer levels, a higher percentage of parameters with normal values was found in the groups without COVID-19 (ward: 29.3%; ICU: 40%) than in the groups with COVID-19 (ward: 12.5%; ICU: 3.8%). The percentage of parameters with altered values, in turn, was higher in patients with COVID-19 (ward: 47.2%; ICU: 46.1%) than in patients without COVID-19 (ward: 28%; ICU: 36.6%). The percentage of parameters with very altered values was also found to be higher in both the groups with COVID-19 (ward: 40.2%; ICU: 50%) and those without COVID-19 (ward: 40.6%; ICU: 23.3%).

Regarding CRP, the percentage of parameters with normal levels was similar between the four groups (both groups with COVID-19 (ward: 17.1%; ICU: 23.5%) and both groups without COVID-19 (ward: 17.3%; ICU: 26.6%). However, the percentage of parameters with altered levels in terms of CRP was higher in the without-COVID-19 groups (ward: 40%; ICU: 20%) than in the with-COVID-19 groups (ward: 27.1%; ICU: 14.7%). Also, among the very altered parameters presented in Table 3, all four groups with COVID-19 (ward: 55.7%; ICU: 61.7%) and without COVID-19 (ward: 42.6%; ICU: 53.3%) had higher biomarker levels.

As for ferritin levels, the percentage of parameters with normal values was higher among groups without COVID-19 (ward: 80%; ICU: 73.3%) than in groups with COVID-19 (ward: 50.7%; ICU: 17.3%). The percentage of parameters with altered values was higher in patients with COVID-19 (ward: 33.3%; ICU: 52.1%) than in patients without COVID-19 (ward: 16%; ICU): 23.3%). The percentage of parameters with very altered values was, also, higher in groups with COVID-19 (ward: 15.8%; ICU: 30.4%) than in groups without COVID-19 (ward: 4%; ICU: 3.3%). The difference between groups was significant regarding D-dimer (*p* = 0.0015) and ferritin (*p* ≤ 0.0001) but not significant regarding CRP (*p* = 0.1233).

The associations between the profile of laboratory biomarkers and the presence of risk characteristics, such as maternal age, gestational age, and presence of comorbidities, were determined (Table 4 and Table 5). Among the analyses performed, only one significant correlation was found, specifically, between the D-dimer value and the gestational trimester of patients with COVID-19 (*p* = 0.0043), and a significant increase in the value of D-dimer levels was observed in relation to the progression of gestational age (Figure 3).

## 4. Discussion

The clinical management of COVID-19 in pregnant women represents a unique challenge for health professionals, not only due to the recent emergence of the new coronavirus but also due to the diversity of physiological and epidemiological factors that can negatively influence the evolution of the disease [13].

Among these factors, increased age is considered a risk factor for COVID-19. In pregnant women, female reproductive age should be considered, which corresponds to 10 to 44 years [14]. In the present study, the age group most affected by COVID-19 was 26 to 34 years old, representing 49.5% (*n* = 60) of the total number of pregnant women with the disease. Among pregnant women without COVID-19, the predominant age group was 16 to 25 years, similar to what was observed among pregnant women with COVID-19 in China and the United States [15,16,17].

The presence of chronic diseases also represents a risk factor for severe COVID-19 in the general population [18,19]. In pregnant women, the existence of these conditions can mean a high-risk pregnancy [4]. A Brazilian epidemiological survey reported maternal deaths resulting from cardiopulmonary complications or multiple organ failure related to COVID-19, the lethality of which was associated with comorbidities such as obesity, diabetes, and cardiovascular disease [4]. In the present study, approximately 21.4% of the pregnant women with COVID-19 had some pre-existing/chronic comorbidity, such as arterial hypertension, asthma, and obesity. Similar studies have also indicated hypertension as the most frequent comorbidity among pregnant women with COVID-19 [20,21].

In addition to chronic diseases, gestational comorbidities are a concern in the face of SARS-CoV-2 infection, since they can contribute to an unfavorable clinical outcome [22]. In the present study, the most frequent comorbidities among pregnant women with COVID-19 were gestational hypertension (GH) and preeclampsia. In the group of pregnant women without COVID-19, these were also the most frequent comorbidities. Several studies have indicated a propensity for pregnant women with COVID-19 to develop preeclampsia, GH, and HELLP syndrome [22,23] and point to an even greater risk of developing these complications among nulliparous women [24]. Another complication reported among patients with COVID-19 is gestational diabetes, a disease that can lead to overweight, excessive tiredness, urinary incontinence, and nausea, among other symptoms [25,26]. In the present study, the frequency of this complication was low, corresponding to 1.6% (*n* = 2) of pregnant women with COVID-19.

Regarding D-dimer, the present study found higher values among pregnant women with COVID-19 compared with pregnant women without the disease, and this difference was statistically significant between the ICU groups. Among maternal deaths, most participants with COVID-19 had elevated D-dimer values, which was also observed in other studies [27,28].

The correlation between D-dimer and gestational age was found to be significant in the present study, specifically in the group of pregnant women with COVID-19. This finding suggests that the presence of SARS-CoV-2 infection in pregnant women in the third trimester influences a higher concentration of D-dimer, in addition to the naturally expected increase during pregnancy [29].

D-dimer levels greater than 1000 ng/mL (2× the reference value) have been associated with a poor prognosis in COVID-19 [12,27], which was observed in the present study. The group without COVID-19 in the ICU, in turn, showed more patients with values within the normal range. It is known that increased D-dimer values suggest the presence of thromboembolic events [11,30]. Klok et al. [31], for example, found in a series of cases in the Netherlands that 31% of pregnant women with COVID-19 in the ICU had some type of venous or arterial thromboembolism, defined as acute pulmonary embolism, ischemic stroke, thrombosis deep vein, or myocardial infarction.

Regarding CRP, several studies have considered it as the most sensitive inflammatory marker for predicting the severity of COVID-19 [29,32,33,34]. In the present study, no difference was found between the CRP levels of pregnant women hospitalized in the ward and ICU with and without COVID-19, similar to what was observed in Poland [34]. Among the maternal deaths recorded in the present study, most pregnant women had increased CRP values, which was also observed in Turkey [32] and in other studies [35].

Regarding the predictive value of ferritin, the present study identified higher concentrations among groups with COVID-19 than in groups without COVID-19. Comparing pregnant women in the ward and ICU with COVID-19, an association was observed between elevated ferritin values and severity of COVID-19. This correlation was also observed in pregnant women in the US and Indonesia, where higher ferritin values were found among patients with severe COVID-19 than in mild/moderate cases [36,37]. In the present study, higher levels of ferritin were also observed in patients who died, similar to what has been reported in other studies conducted in China [38] and in other countries [7].

Several studies have suggested the combined use of biomarkers to obtain a better prognosis in cases of COVID-19. Sahin et al. [32] indicated the use of PCR together with ferritin for greater sensitivity. They also suggested that coagulation markers are not useful for predicting severity during pregnancy, which differs from the results of the present study, since the levels of D-dimer were significantly higher among pregnant women with COVID-19 in the ICU. Alzoughool et al. [39], in turn, suggested the combined use of CRP and D-dimer, proposing that a sixfold increase in both may indicate a greater propensity to develop severe cerebrovascular complications in COVID-19, such as stroke. As a limitation of this study, it should be highlighted that our sample size was relatively small, and it could be underpowering the importance of our statistical associations and null findings for COVID-19 severity among pregnant women in both groups; thus, our findings should be interpreted with caution. Also, due to our limitations, we could not find a correlation for elevated biomarkers being specifically caused by COVID-19, or if other factors or other infections could have been caused these elevated levels.

## 5. Conclusions

The epidemiological and laboratory findings of the present study indicate that being in the age group 26 to 34 years is also suggestive of a higher risk for moderate or severe disease. The most sensitive laboratory biomarkers in the prognosis of cases were ferritin and D-dimer, the levels of which were significantly higher among patients with COVID-19 than in pregnant women without the disease, which was not observed for CRP. Pregnant women in the ICU were the most affected by increases in the levels of ferritin and D-dimer, suggesting an association between high concentrations of these biomarkers and severity of COVID-19 infection.

## Figures and Tables

**Figure 1 ijerph-20-06415-f001:**
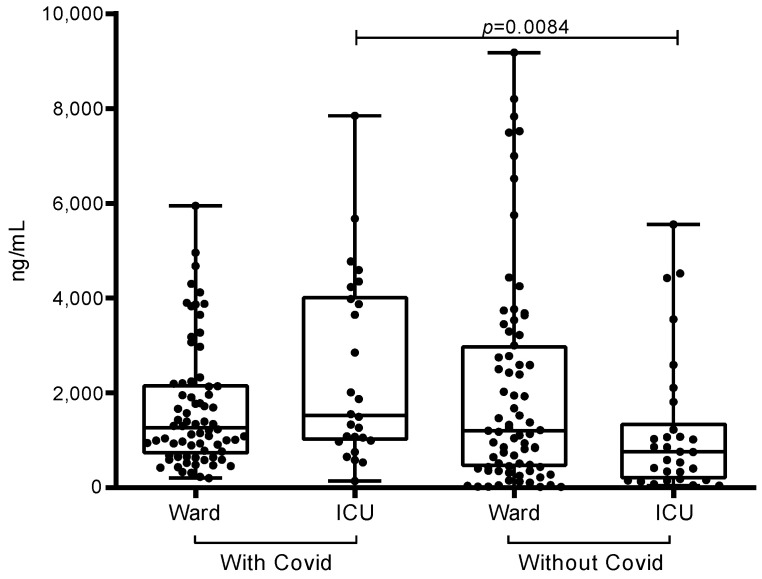
D-dimer profile in pregnant women with and without COVID-19, according to ward and ICU admissions in the city of Belém, Pará, Northern Brazil.

**Figure 2 ijerph-20-06415-f002:**
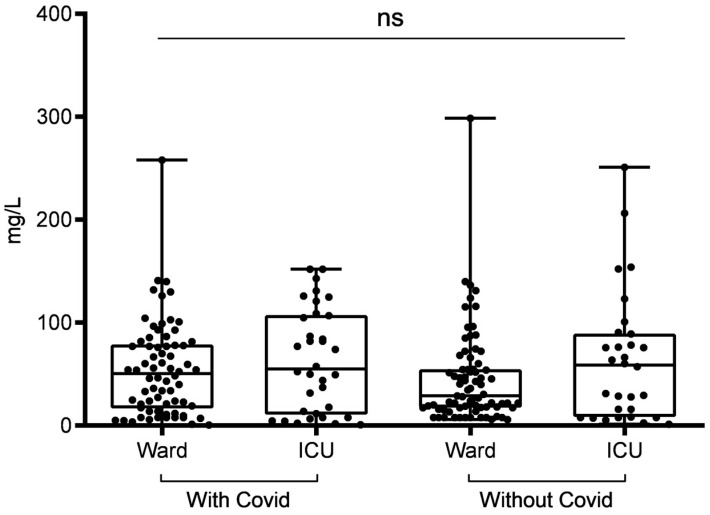
Profile of C-reactive protein in pregnant women with and without COVID-19, according to ward and ICU admission in the city of Belém, Pará, Northern Brazil. ns = not significant.

**Figure 3 ijerph-20-06415-f003:**
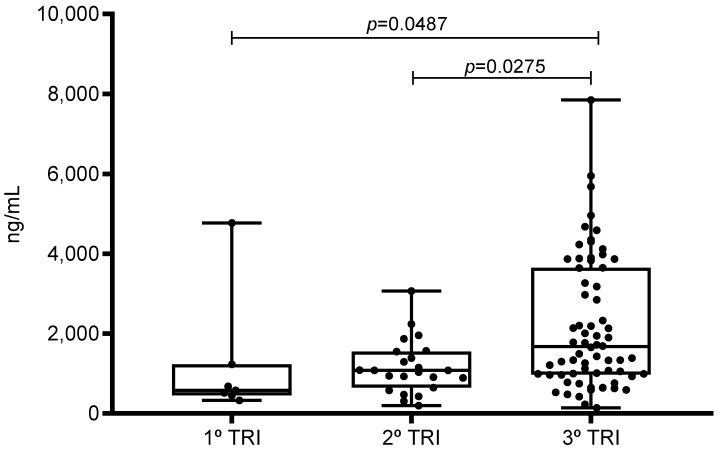
Relationship between D-dimer and gestational trimester in pregnant women with COVID-19 treated in the city of Belém, Pará, northern Brazil. Significant correlations between biomarkers higher levels among pregnant with COVID-19 by trimester.

**Table 1 ijerph-20-06415-t001:** Epidemiological characteristics of pregnant women, classified into two groups according to the presence of SARS-CoV-2 infection: with COVID-19 and without COVID-19.

Characteristics	With COVID-19		Without COVID-19		Total		*p* Value
	*n*	%	*n*	%	*n*	%	
Total	121	53.53	105	46.46	226	100	-
Age							
Mean (SD)	28.44 (6.35)		26.54 (7.52)		27.56 (6.97)		-
Age group							
≤15	3	2.4	3	2.8	6	2.6	
16–25	37	30.5	53	50.4	90	39.8	0.0216 ^c^
26–34	60	49.5	34	32.3	94	41.5	
≥35	21	17.3	15	14.2	36	15.9	
Origin							
Metropolitan region	59	48.7	55	52.3	112	49.5	0.6821 ^a^
Countryside	62	51.2	50	47.6	114	50.4	
Gestational Age							
1st Trimester	7	5.7	8	7.6	15	6.6	
2nd Trimester	27	22.3	19	18.1	46	20.3	0.6689 ^c^
3rd Trimester	87	71.9	78	74.2	165	73.0	
Comorbidities							
Pre-existing	14	11.5	15	14.2	29	12.8	
Gestational	22	18.1	31	29.5	53	23.4	0.0712 ^c^
Pre-existing/Gestational	12	9.9	4	3.8	16	7.0	
Absent	73	60.3	55	52.3	128	56.6	
Pre-existing Comorbidities							
Asthma	6	4.9	6	5.7	12	5.3	0.9643 ^a^
Obesity	4	3.3	1	0.9	5	2.2	0.3758 ^b^
Type II diabetes	1	0.8	0	0.0	1	0.4	1.0000 ^b^
Arterial hypertension	7	5.7	2	1.9	9	3.9	0.1809 ^b^
Acute Renal Failure	3	2.4	0	0.0	3	1.3	0.2504 ^b^
Cardiovascular disease	2	1.6	0	0.0	2	0.8	0.5003 ^b^
Autoimmune Disease	1	0.8	2	1.9	3	1.3	0.5985 ^b^
Infectious diseases	3	2.4	7	6.6	10	4.4	0.1942 ^b^
Other	6	4.9	4	3.8	10	4.4	0.7548 ^b^
Gestational Comorbidities							
Pre-eclampsia	10	8.2	19	18.1	29	12.8	0.0450 ^a^
Gestational Hypertensive	24	19.0	15	14.2	39	17.2	0.3552 ^a^
Gestational diabetes	2	1.6	1	0.9	3	1.3	1.0000 ^b^
Other	0	0.0	2	1.9	2	0.8	0.2147 ^b^
Place of Hospitalization							
Ward	86	71.0	75	71.4	161	71.2	0.9294 ^a^
ICU	35	28.9	30	28.5	65	28.7	
Hospitalization Time—Average (SD)	10.42 (8.34)		9.02 (7.97)		9.02 (7.97)		-
Maternal Outcome							
Medical release	110	90.9	101	96.1	211	93.3	0.1790 ^b^
Decease	11	9.0	4	3.8	15	6.6	

^a^ Chi-square, ^b^ Fisher’s Exact Test, ^c^ G test.

**Table 2 ijerph-20-06415-t002:** Profile of laboratory biomarkers in pregnant women according to four groups: with COVID-19/ward, with COVID-19/ICU, without COVID-19/ward, without COVID-19/ICU.

Biomarkers	With COVID-19		Without COVID-19		*p* Value *	With COVID-19	Without COVID-19	*p* Value **
	Ward	ICU	Ward	ICU				
D-dimer (ng/mL)								
Median	1263	1521	1200	755.5		1316	1024	
(IQR)	(1412.5)	(2942)	(2452.4)	(964)	0.0122	(1686.2)	(2235)	0.0294
Sample	*n* = 72	*n* = 26	*n* = 75	*n* = 30		98	105	
CRP (mg/L)								
Median	50.6	55.05	29	58.75		52.55	31.1	
(IQR)	(60.1)	(94.3)	(37.2)	(75.97)	0.3752	(71.2)	(56.3)	0.2032
Sample	*n* = 70	*n* = 34	*n* = 75	*n* = 30		104	105	
Ferritin (ng/mL)								
Median	140	362.9	79.5	99.1		165.8	84.5	
(IQR)	(235.6)	(352.45)	(78.3)	(136.6)	<0.0001	(278.9)	(83.7)	<0.0001
Sample	*n* = 63	*n* = 23	*n* = 75	*n* = 30		86	105	

* Kruskal–Wallis; ** Mann–Whitney; IQR = Interquartile Range.

**Table 3 ijerph-20-06415-t003:** Levels of laboratory biomarkers between the 4 groups, classified as normal (within the reference), altered (up to 2× the reference value for D-dimer and ferritin and above 2× for CRP) and very altered (greater than 2× the value reference for D-dimer and ferritin and above × for PCR).

	With COVID-19				Without COVID-19				Total with and without COVID-19		*p* Value
	Ward		ICU		Ward		ICU				
	*n*	%	*n*	%	*n*	%	*n*	%	*n*	%	
D-dimer (ng/mL)											
Normal (≤500)	9	12.5	1	3.8	22	29.3	12	40.0	44	21.6	
Altered (501–1500)	34	47.2	12	46.1	21	28.0	11	36.6	78	38.4	0.0015 ^a^
Very altered (>1500)	29	40.2	13	50.0	32	42.6	7	23.3	81	39.9	
CRP											
Normal (≤8.0)	12	17.1	8	23.5	13	17.3	8	26.6	41	19.6	
Altered (9–40)	19	27.1	5	14.7	30	40.0	6	20.0	60	28.7	0.1233 ^b^
Very altered (>40)	39	55.7	21	61.7	32	42.6	16	53.3	108	51.6	
Ferritin											
Normal (≤150)	32	50.7	4	17.3	60	80.0	22	73.3	118	61.7	
Altered (151–450)	21	33.3	12	52.1	12	16.0	7	23.3	52	27.2	<0.0001 ^a^
Very altered (>450)	10	15.8	7	30.4	3	4.0	1	3.3	21	10.9	

^a^ G test; ^b^ Chi-square.

**Table 4 ijerph-20-06415-t004:** Association between laboratory biomarkers and risk characteristics in the group of pregnant women with COVID-19.

Pregnant Women with COVID-19						
Characteristics	D-dimer (ng/mL) Median (IQR)	*p*	CRP (mg/L) Mediana (IQR)	*p*	Ferritin (ng/mL) Mediana (IQR)	*p*
Age	1316 (1500.75)	0.8201 ^b^	52.55 (70.4)	0.7827 ^b^	165.75 (270.32)	0.7817 ^b^
Gestational Age		0.0043 ^a^		0.6108 ^a^		0.4280 ^a^
1st Trimester	580 (469.5)		78 (69.85)		283 (188.2)	
2nd Trimester	1080 (700)		51.3 (38.65)		152 (265.55)	
3rd Trimester	1675 (2662)		48.6 (75.2)		161 (285.25)	
Comorbidities		0.3759 ^a^		0.7892 ^a^		0.2073 ^a^
Yes	1207 (1286)		51.25 (60.4)		175 (265.8)	
No	1339.5 (1914.5)		54.95 (72.7)		140 (267)	

^a^ Kruskal–Wallis; ^b^ Pearson’s correlation.

**Table 5 ijerph-20-06415-t005:** Association between laboratory biomarkers and risk characteristics in the group without COVID-19.

Pregnant Women without COVID-19						
Characteristics	D-dimer (ng/mL) Mediana (IQR)	*p*	CRP (mg/L) Mediana (IQR)	*p*	Ferritin (ng/mL) Mediana (IQR)	*p*
Age	1024 (2234)	0.0678 ^b^	31.1 (56.2)	0.7717 ^b^	84.5 (81.5)	0.4039 ^b^
Gestational age		0.0818 ^a^		0.7034 ^a^		0.4449 ^a^
1st Trimester	390 (498.75)		19.8 (47.37)		141.9 (173.8)	
2nd Trimester	739 (679)		29 (41.7)		110 (89.7)	
3rd Trimester	1205 (2733.75)		35.45 (57.65)		78.8 (71.05)	
Comorbidities		0.8119 ^a^		0.3214 ^a^		0.8294 ^a^
Yes	1054 (2326.20)		28.8 (50.35)		97.7 (72.7)	
No	951 (2135)		43.1 (54)		79.5 (86.6)	

^a^ Kruskal–Wallis; ^b^ Pearson’s correlation.

## Data Availability

All data referred to within this study are available within the manuscript.

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
