# Peer review of "Association of D-Dimer, C-Reactive Protein, and Ferritin with COVID-19 Severity in Pregnant Women: Important Findings of a Cross-Sectional Study in Northern Brazil"

_ijerph, 2023, doi:10.3390/ijerph20146415_

Round 1

Reviewer 1 Report

Researchers in this study evaluated levels of biomarkers that have been associated with COVID-19 severity, D-Dimer, C-Reactive Protein, and Ferritin, among pregnant people hospitalized in Brazil. The authors found elevated levels of D-Dimer and Ferritin among those with confirmed SARS-C0V-2 infection in the ICU, though no associations for CRP. Overall, the paper would benefit from some better organization and some additional thought about how to best present the results. Below are my major comments:

1.      Based on reading the abstract, it seemed the authors compared levels among those with and without COVID-19 diagnosis, but they actually compared them based on level of hospitalization (ward vs. ICU) only. Then, in the conclusion, the authors take three paragraphs to discuss “epidemiological” risk factors associated with COVID-19 severity, even though the abstract and introduction were organized around describing biomarkers. I would suggest focusing more on the biomarker findings, as indicated in the abstract and the body of the rest of the paper, because the findings related to gestational age could be largely influenced by selection bias. Specifically, the women in the study did not necessarily present at the hospital for COVID-19, as evidenced by the fact that there were women hospitalized who tested negative for SARS-CoV-2—all of the participants were tested AFTER hospitalization. Therefore, the findings regarding severity in the third trimester could be related to something else entirely, such as women being more likely to be referred to the hospital in the third trimester vs. other trimesters in general.

a.       Similarly, if only symptomatic cases were tested, isn’t it possible that they had other infections as well?

2.       Introduction—The authors provide a good description of the use of these biomarkers in association with COVID-19 severity, but do not highlight the novelty of their work. It is unclear what this study adds relative to what is already known.

3.       Methods—Generally clear, but I am not sure why the authors limited analyses to ward and ICU and did not compare overall biomarker levels in those with and without COVID-19. The sample size is already small, and it is difficult to draw any real inference from differences between groups when there are only 26-30 in the ICU. For example, Table 2, I would have liked to see two total columns, one for total with COVID-19 and one for total without. The current p-value presentation in this table is a bit meaningless, as it only helps determine if there is a statistically significant difference between ANY of the four groups. There is a missing relevant p-value, which is for the comparison between TOTAL with and without COVID-19. I am not sure how useful it is to compare ward vs. ICU among COVID groups, since the authors do not spend much time contextualizing these implications in the discussion.

4.       The Results section is highly duplicative—there is no reason to spend 2 paragraphs writing about descriptive statistics of the sample when the information is all in Table 1. It was unclear if the text of the results was providing additional information beyond what was in the tables/figures. I would expect some text to highlight results, of course, but there should be less overlap between text and tables/figures. There should also be some thought into finding a more useful way of presenting the results. The notations in the box plots (e.g., Figure 3) of statistical significance are not intuitive or helpful. As mentioned in my previous comment, I do not see the utility in comparing ward to ICU within COVID, and would rather see a side-by-side comparison of ward with COVID vs. ward without COVID. Maybe the way it is presented currently is better for the message of the paper, but the message is not clear to me—from the way the paper was written, I got the impression that the authors were going to make overall comparisons between COVID vs. none.

5.       Results—starting line 221, authors mention four additional groups based on serum levels of biomarkers. These results come out of nowhere and the rationale for these levels should be explained in the Methods (especially when later in the discussion 2x normal limit is mentioned, so where does 3x come from?).

6.       “Correlations” between biomarkers and “risk characteristics.” (starting line 251) Again, based on the information presented before this point, I do not see the relevance of this analysis, and the use of the word “correlation” throughout it incorrect. The authors are studying associations, whereas a correlation describes a specific association between continuous variables. Conclusion about third trimester—in addition to my concerns above—there is huge variation in D-dimer in the third trimester for both those with and without COVID, so I’m not sure what conclusions can really be drawn about the third trimester (vs. the other 2), particularly when D-dimer is already higher in the third trimester, naturally.  

7.       General conclusions—though the authors did not find an association between CRP and COVID-19 severity, they are underpowered to be able to detect a truly null association. Rather than concluding there was no association, the authors should be clear that they had a small sample size and may be underpowered to detect an association, so results should be interpreted with caution. It should also be acknowledged that the authors could not adjust analyses for any conditions that may contribute to both severity of COVID-19 and biomarker levels (e.g., other infection, respiratory conditions, etc). I see no strengths or limitations mentioned at all.

8.       Minor—Could authors use a different term than “pregnancy specific hypertension syndrome?” It is unclear what this would mean for an international audience (is it just hypertensive disorders of pregnancy? Gestational hypertension?)—it doesn’t seem like this terminology is used internationally. Line 314-315—“Among maternal deaths . . .” This information does not appear in the methods or results. Authors describe the study hospital as a “reference” hospital. Please clarify what this means—is it a hospital where people are frequently referred from other places?

English language was mostly fine, though I think there may be some translation issues with a couple of terms, as noted in my final comment--I've never seen Pregnancy-specific hypertensive syndromes referenced outside of Brazilian publications.

Author Response

We appreciate the Reviewer 1's availability of time for improving the manuscript. All suggestions were accepted, and modifications were made according to the description below:

  1. Based on reading the abstract, it seemed the authors compared levels among those with and without COVID-19 diagnosis, but they actually compared them based on level of hospitalization (ward vs. ICU) only. Then, in the conclusion, the authors take three paragraphs to discuss “epidemiological” risk factors associated with COVID-19 severity, even though the abstract and introduction were organized around describing biomarkers. I would suggest focusing more on the biomarker findings, as indicated in the abstract and the body of the rest of the paper, because the findings related to gestational age could be largely influenced by selection bias. Specifically, the women in the study did not necessarily present at the hospital for COVID-19, as evidenced by the fact that there were women hospitalized who tested negative for SARS-CoV-2—all of the participants were tested AFTER hospitalization. Therefore, the findings regarding severity in the third trimester could be related to something else entirely, such as women being more likely to be referred to the hospital in the third trimester vs. other trimesters in general.

R: We appreciate the important observation made by the reviewer regarding the risk factors associated with COVID-19, and we agree that it is not the main objective of the study. Therefore, we have decided to modify the text in the discussion and conclusions accordingly. The deleted sentences was:

Lines 293-296– Results: “Among the analyzes performed, only one significant correlation was found, specifically, between the D-dimer value and the gestational trimester of patients with COVID-19 (p = 0.0043) and a significant increase in the value of D-dimer was observed in relation to the progression of gestational age (Figure 4).”

Lines 320-325 - Discussion - “Some studies indicate that the 3rd trimester of pregnancy represents a higher risk for severe COVID-19 [13,17]. In the present study, most pregnant women with COVID-19 were in the 3rd trimester, which reinforces the idea that this period represents the majority of hospitalizations. In addition, all 15 maternal deaths recorded in this study occurred among pregnant women in the 3rd trimester, of which 11 had confirmed infection with SARS-CoV-2.”

Lines 358-360: This finding suggests that the presence of SARS-CoV-2 infection in pregnant women in the 3rd trimester influences the higher concentration of D-dimer, in addition to the naturally expected increase during pregnancy [29].

The conclusion has been reworked, and the following sentence has been removed:

Lines 397-399: “...the 3rd trimester as the gestational period with the highest risk of severity for COVID-19 in pregnant women.”

Therefore, the conclusion is as follows:

Lines 397-405: “The epidemiological and laboratory findings of the present study indicate the age group of 26 to 34 years can also be suggested as a higher risk for the detection of moderate or severe disease. The most sensitive laboratory biomarkers in the prognosis of cases were ferritin and D-dimer, which were significantly higher among patients with COVID-19 than in pregnant women without the disease, which was not observed for CRP. Pregnant women in the ICU were the most affected by the increase of ferritin and D-dimer, suggesting an association between high concentrations of these biomarkers and severity in COVID-19.

The epidemiological and laboratory findings of the present study indicate the age group of 26 to 34 years can also be suggested as a higher risk for the detection of moderate or severe disease. The most sensitive laboratory biomarkers in the prognosis of cases were ferritin and D-dimer, which were significantly higher among patients with COVID-19 than in pregnant women without the disease, which was not observed for CRP. Pregnant women in the ICU were the most affected by the increase of ferritin and D-dimer, suggesting an association between high concentrations of these biomarkers and severity in COVID-19.

  1. Similarly, if only symptomatic cases were tested, isn’t it possible that they had other infections as well?

R: As all pregnant women, whether with or without COVID-19, were hospitalized, either in the ward or in the ICU, it is possible that various other causes, in addition to COVID-19, were involved in the need for hospitalization. This information is also included in the discussion as described below:

Lines 337-339: “In the present study, the most frequent comorbidities among pregnant women were gestational hypertension and pre-eclampsia.”

  1. Introduction—The authors provide a good description of the use of these biomarkers in association with COVID-19 severity, but do not highlight the novelty of their work. It is unclear what this study adds relative to what is already known.

R: The distinctive feature of this study lies in the investigated population. It is known that pregnancy naturally increases these markers, especially D-dimer. Our interest was to determine if COVID-19 promoted an increase beyond the expected range, and this was observed for D-dimer and ferritin. Another important finding regarding ferritin was the significant difference observed among almost all investigated groups. This allows us to suggest that ferritin is a highly effective inflammatory marker in monitoring cases of pregnant women with COVID-19.

  1. Methods—Generally clear, but I am not sure why the authors limited analyses to ward and ICU and did not compare overall biomarker levels in those with and without COVID-19. The sample size is already small, and it is difficult to draw any real inference from differences between groups when there are only 26-30 in the ICU. For example, Table 2, I would have liked to see two total columns, one for total with COVID-19 and one for total without. The current p-value presentation in this table is a bit meaningless, as it only helps determine if there is a statistically significant difference between ANY of the four groups. There is a missing relevant p-value, which is for the comparison between TOTAL with and without COVID-19. I am not sure how useful it is to compare ward vs. ICU among COVID groups, since the authors do not spend much time contextualizing these implications in the discussion.

R: We appreciate the reviewer's observation, and we have added the column with statistical analysis for the total cases with and without COVID-19 to Table 2, as well as the cases of COVID-19 in the ward and ICU, as suggested.

  1. The Results section is highly duplicative—there is no reason to spend 2 paragraphs writing about descriptive statistics of the sample when the information is all in Table 1. It was unclear if the text of the results was providing additional information beyond what was in the tables/figures. I would expect some text to highlight results, of course, but there should be less overlap between text and tables/figures. There should also be some thought into finding a more useful way of presenting the results. The notations in the box plots (e.g., Figure 3) of statistical significance are not intuitive or helpful. As mentioned in my previous comment, I do not see the utility in comparing ward to ICU within COVID, and would rather see a side-by-side comparison of ward with COVID vs. ward without COVID. Maybe the way it is presented currently is better for the message of the paper, but the message is not clear to me—from the way the paper was written, I got the impression that the authors were going to make overall comparisons between COVID vs. none.

R: We appreciate your suggestion and our comparison between a common ward and an intensive ward such as ICU is to verify whether symptomatic clinical data associated with the severity of COVID-19 infection during pregnancy would influence biomarker levels differently. That is, this was the first study in the literature to investigate whether clinical conditions, gestational age and hospitalization conditions would be relevant enough to change the levels of these biomarkers and thus there could be complications for the pregnant woman and newborns, consequently there is a need to differentiated treatment in pregnant women with high biomarkers depending on the hospital wing where they were.

We have removed all duplicated data. It was not easy to find a way to demonstrate the comparison results between cases with COVID-19 and without COVID-19 within the ward and ICU. In Figures 1, 2, and 3, we compare both within and between groups. For example, in Figure 3, we compare the ward and ICU with COVID-19 (p=0.0196), the ward with COVID-19 and the ward without COVID-19 (p=0.0052), the ICU with COVID-19 and the ward without COVID-19 (p<0.0001), and the ICU with COVID-19 and the ICU without COVID-19 (p<0.0001). In this way, we have made all possible comparisons among the four groups of pregnant women evaluated. Only statistically significant differences were shown in the Figures. To improve the reader's understanding of these results, we have modified the text as described below:

Lines 207-213: "Regarding the biomarker profile, there was a significant difference in D-dimer (p=0.0294) and Ferritin (p<0.0001) between the groups (with and without COVID) (Table 2). However, the difference observed in CRP between the groups was not statistically significant (p=0.3752) (Figure 2). When we further divided these groups based on the location of hospitalization, we found that only the ICU patients with COVID-19 had significantly higher D-dimer values compared to ICU patients without COVID-19 (1521; 2942 vs 755.5; 864, p=0.0084) (Figure 2), while other comparisons did not show significant differences."

Lines 216-224: “Until our study, there has not been no data confirmation about COVID-19 biomarkers levels among pregnant women, in both ward or UCI or even a comparation between these two hospital areas. In our results, we demonstrated that SARS-CoV-2 infected pregnant women in UCI may have a greater predisposition to D-dimer, CRP and ferritin higher levels associated to a severe acute respiratory syndrome symptoms, therefore those women in SARS-CoV-2 infected group need admission to UCI, mainly, due to the physiological changes that occur during pregnancy, women might be more vulnerable to higher D-dimer, CRP and ferritin levels and then severe respiratory infections and associated complications.Parte superior do formulárioParte inferior do formulário

  1. Results—starting line 221, authors mention four additional groups based on serum levels of biomarkers. These results come out of nowhere and the rationale for these levels should be explained in the Methods (especially when later in the discussion 2x normal limit is mentioned, so where does 3x come from?).

R: We appreciate the important observation made by the reviewer. We did not find in the literature any parameter for comparing the serum levels of the evaluated biomarkers in the study with the prognosis of COVID-19 or the severity of the disease. Therefore, we decided to establish cutoff values for each of the markers in order to attempt to establish an association between these serum levels and the clinical outcome of COVID-19 in the population of pregnant women. We have included a paragraph in the methods section that explains this association, as described below:

Lines 133-137: "To compare the biomarkers among the four groups of pregnant women in the study and their association with the severity of COVID-19, the serum levels were classified as follows: normal (within the reference range), altered (up to 3 times the reference value for D-dimer and ferritin, and up to 5 times for CRP), and highly altered (above 3 times the reference value for D-dimer and ferritin, and above 5 times for CRP)."

  1. “Correlations” between biomarkers and “risk characteristics.” (starting line 251) Again, based on the information presented before this point, I do not see the relevance of this analysis, and the use of the word “correlation” throughout it incorrect. The authors are studying associations, whereas a correlation describes a specific association between continuous variables. Conclusion about third trimester—in addition to my concerns above—there is huge variation in D-dimer in the third trimester for both those with and without COVID, so I’m not sure what conclusions can really be drawn about the third trimester (vs. the other 2), particularly when D-dimer is already higher in the third trimester, naturally.  

R: The reviewer is correct, and we apologize for the mistake regarding the term "correlation." In the manuscript and the captions of Tables 4 and 5, the term "correlation" has been replaced with "association." Regarding the discussion, we have removed the information regarding the variation of D-dimer in the third trimester of pregnancy, and we have also modified the conclusions to focus solely on the studied biomarkers.

  1. General conclusions—though the authors did not find an association between CRP and COVID-19 severity, they are underpowered to be able to detect a truly null association. Rather than concluding there was no association, the authors should be clear that they had a small sample size and may be underpowered to detect an association, so results should be interpreted with caution. It should also be acknowledged that the authors could not adjust analyses for any conditions that may contribute to both severity of COVID-19 and biomarker levels (e.g., other infection, respiratory conditions, etc). I see no strengths or limitations mentioned at all.

R: We appreciate your suggestion and the conclusion was carefully revised. We added a paragraph highlighting some limitations of the study, as described below:

Lines 391-395: “As a limitation of this study, it should be highlighted that our sample size was relatively small, and it could impact in our statistical association’s importance, such as, CRP and COVID-19 severity among pregnant women in both groups. Also, due to our limitations we could not correlate medical conditions like secondary infections.”

  1. Minor—Could authors use a different term than “pregnancy specific hypertension syndrome?” It is unclear what this would mean for an international audience (is it just hypertensive disorders of pregnancy? Gestational hypertension?)—it doesn’t seem like this terminology is used internationally. Line 314-315—“Among maternal deaths . . .” This information does not appear in the methods or results. Authors describe the study hospital as a “reference” hospital. Please clarify what this means—is it a hospital where people are frequently referred from other places?

R: We appreciate the observation. The term "pregnancy specific hypertension syndrome" has been replaced with "Gestational hypertension" as it is more widely recognized worldwide.

Regarding the information "Among maternal deaths..." (line ….), it has been removed from the discussion as it was indeed not mentioned earlier. The excluded sentence was:

Lines 354-356: "Among maternal deaths, most participants with COVID-19 had elevated D-dimer values, which was also observed in other studies [27,28]."

We also decided to delete the sentence referring to deaths in the discussion. The excluded sentence was:

Lines 372-374: Among the maternal deaths recorded in the present study, most pregnant women had increased CRP values, which was also observed in Turkey [32] and in other studies [35].

However, we only leave a description of the deaths that occurred in the results, as described below:

Lines 165-167: As for the maternal outcome, 90.9% (n= 110) of pregnant women with COVID-19 were discharged and 9.1% (n= 11) died, while the participants without COVID-19, 96.1% (n = 101) were discharged and 3.8% (n = 4) died.

Regarding the term "reference hospital" mentioned in the methodology, it was used because the majority of pregnant women in the state of Pará, who have pregnancy complications or high-risk pregnancies, are directed to FSCMP. To ensure better understanding for all readers, we have modified the description of the hospital as follows:

Line 92-94: "The FSCMP hospital was the facility where pregnant women from the state of Pará, who had clinical complications requiring hospitalization, were referred to during the COVID-19 pandemic."

Reviewer 2 Report

Comments

Line 121: “The tests were requested during hospital admission by the physician in charge” Specify the time point when the tests were performed: on admission or later or you have  used the maximal measured levels?

 Line 176 – table 2 - What was the reason not to perform markers in all patient with COVID (as it was the case in group of patients without COVID)? There were 121 pregnant patients with COVID, and D dimer levels were measured in 98, etc. The same comment for table 3.

 Line 181: COVID-19/ICU, without COVID-19/ward and without COVID-19/UTI (Table 2).” Explain or correct “UTI”.

 Line 247:  Table 3 - “altered (up to 3x the reference value for D-dimer and ferritin, and above 5x for CRP)” Above 5x for CRP or up to 5x for CRP (as it was stated in line 223)?

Author Response

We appreciate the Reviewer's availability of time for the important contributions to improving the manuscript. All suggestions were accepted, and the corrections are described below:

Line 121: “The tests were requested during hospital admission by the physician in charge” Specify the time point when the tests were performed: on admission or later or you have  used the maximal measured levels?

R: We appreciate the observation. All results regarding the biomarkers were obtained upon admission of the study participants to the hospital. To eliminate any doubt regarding this information, we have included the following sentence:

Lines 132-133: All results regarding the studied biomarkers were obtained at the time of participants' hospitalization.

Line 176 – table 2 - What was the reason not to perform markers in all patient with COVID (as it was the case in group of patients without COVID)? There were 121 pregnant patients with COVID, and D dimer levels were measured in 98, etc. The same comment for table 3.

R: Both tables were carefully revised.

 Line 181: “COVID-19/ICU, without COVID-19/ward and without COVID-19/UTI (Table 2).” Explain or correct “UTI”.

R: We appreciate the reviewer's observation, and we have added the column with statistical analysis for the total cases with and without COVID-19 to Table 2. Please note that not all participants had results for all three biomarkers, as the request for all three tests is not mandatory upon admission of pregnant women to the hospital. As a result, the sample size varies for each biomarker. The acronym UTI was deleted. This information has been added to the results section as described below:

Lines 214-217: “It is worth noting that not all study participants had results for all three evaluated biomarkers, as the admission tests were requested by physicians based on the clinical characteristics of each patient.”

Line 247:  Table 3 - “altered (up to 3x the reference value for D-dimer and ferritin, and above 5x for CRP)” Above 5x for CRP or up to 5x for CRP (as it was stated in line 223)?

R: We appreciate the important observation made by the reviewer. In order to enhance the understanding of the evaluated laboratory parameters, we have added the following information to the methodology:

Lines 134-138: "To compare the biomarkers among the four groups of pregnant women in the study and their association with the severity of COVID-19, the serum levels were classified as follows: normal (within the reference range), altered (up to 3 times the reference value for D-dimer and ferritin, and up to 5 times for CRP), and highly altered (above 3 times the reference value for D-dimer and ferritin, and above 5 times for CRP)."

Reviewer 3 Report

Respected Editor,

Thank you for the chance to assess the manuscript titled "Association of D-dimer, C-reactive protein, and Ferritin with COVID-19 severity in pregnant women: important findings of a cross-sectional Study in Northern Brazil."

 The researchers carried out a cross-sectional observational investigation aimed at examining the association between these markers and the disease severity in pregnant women. Each investigation pertaining to the correlation or influence of COVID-19 on pregnancy holds great importance for the obstetric community, and the current study is not an exception. 

The manuscript's introduction adequately establishes the necessary background, and the references cited are pertinent to the research. 

Even so, certain sections of the manuscript necessitate significant revisions, and the authors must address several questions to enhance the scientific soundness of the draft. 

  1. Materials and methods. What were the criteria for the patients' administration to the ICU and the ward in both groups?
  2. Statistical analysis and the results. Why is there a lack of comparison of the mentioned markers between COVID-19-positive patients admitted to the ICU and the ward? The authors compared only the serum values of ferritin between these groups. 
  3. D-dimer is elevated in pregnancy. How can authors distinguish the impact of COVID-19 on D-dimer elevation and the impact of the pregnancy itself? This remark is of great significance
  4. How do the authors explain the significantly higher percentage of preeclampsia in the non-COVID group? The authors state clearly that the gestation comorbidities were more frequent in COVID-19-positive pregnant women in other studies. 
  5. Even though the number of subjects is too small to draw any conclusion, adding the table with the deceased patient's characteristics would improve the draft.
  6. D-dimer naturally increases with gestational age. How do the authors explain the correlation between this correlation in COVID-19 pregnant women?
  7. Arguably, the manuscript's highlight was the correlation between serum ferritin and the severity of the COVID-19 disease. The authors should provide more details regarding the serum ferritin in pregnancy and the possible explanation of its elevation in the more severe disease. 

Minor remarks:

  1. Introduction, lines 42 to 44: these variables present the important epidemiological determinants of? The sentence seems unfinished. 
  2. Results, line 162: The authors should change the abbreviation SHSP.
  3. Results, line 169: The authors should change the sentence. It would be better to say that the eclampsia was more frequent in the patients without COVID-19, rather than "patients with preeclampsia were more frequent in the group without COVID-19 than in the group with the COVID-19"
  4. Discussion, lines 292 and 293: the presence of these comorbidities means a high-risk pregnancy. The authors should exclude the "can mean". 
  5. Discussion, line 303: The authors should change the term "pregnancy-specific hypertensive syndrome (PSHS)" to "hypertensive disorders of pregnancy (HDPs)". 

Author Response

We appreciate the Reviewer 3's availability of time for improving the manuscript. All suggestions were accepted, and modifications were made according to the description below:

  1. Materials and methods. What were the criteria for the patients' administration to the ICU and the ward in both groups?

R: The criteria were specifically clinical, regardless of whether the pregnant women had a confirmed diagnosis of COVID-19 or not.

2. Statistical analysis and the results. Why is there a lack of comparison of the mentioned markers between COVID-19-positive patients admitted to the ICU and the ward? The authors compared only the serum values of ferritin between these groups. 

R: The comparison between the serum levels of the studied biomarkers is shown in figures 1, 2, and 3. We have added a paragraph in the results section regarding the comparison between the groups with and without COVID-19, as described below:

“Regarding the biomarker profile, there was a significant difference in D-dimer (p=0.0294) and Ferritin (p<0.0001) between the groups (with and without COVID) (Table 2). However, the difference observed between groups in CRP was not statistically sig-nificant (p=0.3752) (Figure 2). When we further divided these groups based on the location of hospitalization, we found that only the ICU patients with COVID-19 had significantly higher D-dimer values compared to ICU patients without COVID-19 (1521; 2942 vs 755.5; 864, p=0.0084) (Figure 2), while other comparisons did not show significant differences.”

3. D-dimer is elevated in pregnancy. How can authors distinguish the impact of COVID-19 on D-dimer elevation and the impact of the pregnancy itself? This remark is of great significance.

R: The reviewer's observation that D-dimer is naturally elevated during pregnancy is of great importance. The statistical analysis demonstrates that both D-dimer and ferritin show significantly higher values in the group of pregnant women with COVID-19 compared to the group of pregnant women without COVID-19. We understand that if COVID-19 had no influence on the levels of these two markers, there would be no statistical significance.

4. How do the authors explain the significantly higher percentage of preeclampsia in the non-COVID group? The authors state clearly that the gestation comorbidities were more frequent in COVID-19-positive pregnant women in other studies. 

R: This point is very important, and we appreciate the comment. Although some studies have made an association between COVID-19 and pre-eclampsia, our study suggests that the onset of pre-eclampsia does not appear to be influenced by COVID-19. Therefore, we have modified the text for better reader comprehension as described below:

Lines 282-292: “In the present study, the most frequent comorbidities among pregnant women were gestational hypertension and pre-eclampsia. Several studies have indicated a propensity for pregnant women with COVID-19 to develop pre-eclampsia and gestational hypertension [22,23] and point to an even greater risk among nulliparous women of develop these complications [24]. Another complication reported among patients with COVID-19 is gestational diabetes, a disease that can lead to overweight, excessive tiredness, urinary incontinence, nausea, among other symptoms [25,26]. In the present study, it was observed that the occurrence of gestational hypertension was not different between pregnant women with and without COVID-19, but the prevalence of pre-eclampsia was significantly higher in the group of pregnant women without COVID-19. This suggests that COVID-19 does not have an influence on the onset of this clinical condition.

5. Even though the number of subjects is too small to draw any conclusion, adding the table with the deceased patient's characteristics would improve the draft.

R: We appreciate the observation, but we justify the absence of this information in the tables, due to the small number of women who died in the period studied, which would make statistical analysis difficult. Perhaps this can be considered a limitation of the study.

6. D-dimer naturally increases with gestational age. How do the authors explain the correlation between this correlation in COVID-19 pregnant women?

R: The correlation between D-dimer and gestational age was found to be significant only in the COVID-19 group (p = 0.0043), but not in the control group (p = 0.0818). This finding suggests that the presence of SARS-CoV-2 infection in pregnant women in the third trimester influences the higher concentration of D-dimer, in addition to the naturally expected increase during pregnancy.

7. Arguably, the manuscript's highlight was the correlation between serum ferritin and the severity of the COVID-19 disease. The authors should provide more details regarding the serum ferritin in pregnancy and the possible explanation of its elevation in the more severe disease. 

R: We appreciate the reviewer's important observation. The literature demonstrates that high levels of ferritin during pregnancy can predispose to adverse maternal and fetal outcomes, such as gestational diabetes mellitus. On the other hand, other studies have shown that COVID-19 patients with elevated levels of ferritin generally exhibit more severe disease courses, requiring ICU hospitalization. Therefore, it is natural to expect that pregnant women with COVID-19 may have even higher levels of ferritin, especially in cases of more severe COVID-19. To make this information clearer for the reader, we have added the following sentence to the manuscript:

Lines 318-325: “Other studies demonstrate a relationship between high ferritin concentrations and the risk of adverse maternal and fetal pregnancy outcomes such as gestational diabetes mellitus in women without COVID-19 (Cheng et al, 2020; Yang et al, 2022). In addition, general patients with COVID-19 who have one or more comorbidities, including diabetes, thrombotic complication and cancer, had significantly higher levels of ferritin and its level been associated with supportive intensive care, including transfer to ICU and mechanical ventilation (Kong et al, 2023). These factors added together could increase the likelihood of major complications during pregnancy in women with COVID-19.”

Minor remarks:

1. Introduction, lines 42 to 44: these variables present the important epidemiological determinants of? The sentence seems unfinished. 

R: The sentence was completed and the new text reads as follows:

“According to the Brazilian Observatory of COVID-19 (OOBr), race (brown or white), age (20-34 years), gestation period (3rd gestational trimester) and area of residence (zone urban) are important epidemiological determinants for monitoring of pregnant and postpartum women”

2. Results, line 162: The authors should change the abbreviation SHSP.

R: The acronym SHSP was replaced by gestational hypertension.

3. Results, line 169: The authors should change the sentence. It would be better to say that the eclampsia was more frequent in the patients without COVID-19, rather than "patients with preeclampsia were more frequent in the group without COVID-19 than in the group with the COVID-19"

R: The manuscript has been modified. New text: “Regarding gestational comorbidities, pre-eclampsia was more frequent in the group without COVID-19 than in the group with COVID-19 (8.2% vs 18.1%, p=0.0450)”.

4. Discussion, lines 292 and 293: the presence of these comorbidities means a high-risk pregnancy. The authors should exclude the "can mean". 

R: The manuscript has been modified. New text: “In pregnant women, the existence of these conditions means a high-risk pregnancy [4]”.

5. Discussion, line 303: The authors should change the term "pregnancy-specific hypertensive syndrome (PSHS)" to "hypertensive disorders of pregnancy (HDPs)". 

R: The term "pregnancy-specific hypertensive syndrome (PSHS) was replaced by gestational hypertension.

Round 2

Reviewer 1 Report

In general, the authors have adequately revised the manuscript, but there are a couple of questions that are outstanding.

1.       Authors respond to the comment “If only symptomatic cases were tested, isn’t it possible that they had other infections as well?” With an addition to the discussion about the frequency of comorbidities (gestational hypertension and preeclampsia), which was not my concern. The concern was that because COVID-19 infection presents with a variety of symptoms, it is possible that participants had other acute infections in addition to COVID-19. I note that the authors added the following line “Also, due to our limitations we could not correlate medical conditions like secondary infections,” which is not quite correct. The limitation is that it is unknown if the elevated biomarkers are specifically due to COVID-19, or if they could have been caused by additional or other infections.

2.       Addition of lines 216-224—needs to be checked for translation issues, but some of the argument doesn’t make sense in the last portion:

“women might be more vulnerable to higher D-dimer, CRP and ferritin levels and then severe respiratory infections and associated complications.”

I think the authors are saying that pregnant women are more vulnerable to infection and more likely to display a higher inflammatory response, exemplified by higher levels of D-dimer, etc. The way it is written makes it sound like the biomarkers are elevated first and then the infection occurs, which doesn’t really make sense from a biological perspective. The authors make a good point that it is important to compare these biomarkers among those in the ICU with and without COVID-19 to determine if there is something specific about being in the ICU alone that may influence D-Dimer, CRP, and ferritin. However, the point is a bit lost in this paragraph, so would be good to see it clarified.

3.       Biomarkers in 4 additional groups: “normal (within the reference range), altered (up to 3 times the reference value for D-dimer and ferritin, and up to 5 times for CRP), and highly altered (above 3 times the reference value for D-dimer and ferritin, and above 5 times for CRP).”
Thank you for describing this in the Methods section, but there needs to be some sort of rationale for categorizing them in this manner. I understand that there may not be existing literature, but there has to be a reason the authors chose 3 and 5 times higher? Do they represent the top 25% and top 10% of values? Please clarify where these numbers come from.

4.       Figure 3—all of the comparisons are significant, so it would be more useful to just include a footnote that makes that point instead of listing p-values for each of the comparisons.

5.       Minor—gestational age is broken into trimesters, not quarters (1st, 2nd, 3rd trimester).

6.       Thank you for adding the overall totals with and without COVID-19 in Table 2. Please be sure to clarify in those last columns that they are the total with and without COVID-19.

In general, the authors have adequately revised the manuscript, but there are a couple of questions that are outstanding.

1.       Authors respond to the comment “If only symptomatic cases were tested, isn’t it possible that they had other infections as well?” With an addition to the discussion about the frequency of comorbidities (gestational hypertension and preeclampsia), which was not my concern. The concern was that because COVID-19 infection presents with a variety of symptoms, it is possible that participants had other acute infections in addition to COVID-19. I note that the authors added the following line “Also, due to our limitations we could not correlate medical conditions like secondary infections,” which is not quite correct. The limitation is that it is unknown if the elevated biomarkers are specifically due to COVID-19, or if they could have been caused by additional or other infections.

2.       Addition of lines 216-224—needs to be checked for translation issues, but some of the argument doesn’t make sense in the last portion:

“women might be more vulnerable to higher D-dimer, CRP and ferritin levels and then severe respiratory infections and associated complications.”

I think the authors are saying that pregnant women are more vulnerable to infection and more likely to display a higher inflammatory response, exemplified by higher levels of D-dimer, etc. The way it is written makes it sound like the biomarkers are elevated first and then the infection occurs, which doesn’t really make sense from a biological perspective. The authors make a good point that it is important to compare these biomarkers among those in the ICU with and without COVID-19 to determine if there is something specific about being in the ICU alone that may influence D-Dimer, CRP, and ferritin. However, the point is a bit lost in this paragraph, so would be good to see it clarified.

3.       Biomarkers in 4 additional groups: “normal (within the reference range), altered (up to 3 times the reference value for D-dimer and ferritin, and up to 5 times for CRP), and highly altered (above 3 times the reference value for D-dimer and ferritin, and above 5 times for CRP).”
Thank you for describing this in the Methods section, but there needs to be some sort of rationale for categorizing them in this manner. I understand that there may not be existing literature, but there has to be a reason the authors chose 3 and 5 times higher? Do they represent the top 25% and top 10% of values? Please clarify where these numbers come from.

4.       Figure 3—all of the comparisons are significant, so it would be more useful to just include a footnote that makes that point instead of listing p-values for each of the comparisons.

5.       Minor—gestational age is broken into trimesters, not quarters (1st, 2nd, 3rd trimester).

6.       Thank you for adding the overall totals with and without COVID-19 in Table 2. Please be sure to clarify in those last columns that they are the total with and without COVID-19.

7. Please explicitly point out in the limitations that the study is underpowered and null findings for CRP should be interpreted with caution. That point is not clear in the revision.

Some of the additions need to be reviewed carefully for English language translation issues, most notably in the revision I note in my comments (#2).

Author Response

Reply to reviewer #1

1. Concern of the reviewer • Discussion comment - Authors respond to the comment “If only symptomatic cases were tested, isn’t it possible that they had other infections as well?” With an addition to the discussion about the frequency of comorbidities (gestational hypertension and preeclampsia), which was not my concern. The concern was that because COVID-19 infection presents with a variety of symptoms, it is possible that participants had other acute infections in addition to COVID-19. I note that the authors added the following line “Also, due to our limitations we could not correlate medical conditions like secondary infections,” which is not quite correct. The limitation is that it is unknown if the elevated biomarkers are specifically due to COVID-19, or if they could have been caused by additional or other infections. Our response:               Dear Reviewer #1, we appreciate your suggestion and the limitations paragraph was revised. Revised text:Page 10, lines 336-342, “As a limitation of this study, it should be highlighted that our sample size was relatively small, and it could underpowering our statistical associations importance and null findings for and COVID-19 severity among pregnant women in both groups, which it should be interpreted with caution. Also, due to our limitations we could not correlate elevated biomarkers are, specifically, caused by COVID-19, or if other factor or other infection could have been caused it.” 2. Concern of the reviewer • Minor comment - Addition of lines 216-224—needs to be checked for translation issues, but some of the argument doesn’t make sense in the last portion: Our response:               Dear Reviewer #1, we appreciate your suggestion and the results paragraph was revised. Revised text:Page 7, lines 213-227, “Regarding CRP normal levels parameters, they were similar between four groups (both groups with COVID-19 (ward: 17.1%; ICU: 23.5%) and both groups without COVID-19 (ward: 17.3%; ICU: 26.6%). Although Regarding CRP altered levels parameters, they were higher in the without COVID-19 groups (ward: 40%; ICU: 20%) than in with COVID-19 groups (ward: 27.1%; ICU: 14.7%). Also, among very altered parameters presented in table 3, all four groups with COVID-19 (ward: 55.7%; ICU: 61.7%) and without COVID-19 (ward: 42.6%; ICU: 53.3%) had higher biomarkers levels. As for ferritin levels, normal values parameters were higher among groups without COVID-19 (ward: 80%; ICU: 73.3%) than in groups with COVID-19 (ward: 50.7%; ICU: 17.3%). The altered values parameters were higher in patients with COVID-19 (ward: 33.3%; ICU: 52.1%) than in patients without COVID-19 (ward: 16%; ICU): 23.3%). Very altered values parameters were, also, higher in groups with COVID-19 (Ward: 15.8%; ICU: 30.4%) than in groups without COVID-19 (Ward: 4%; ICU: 3.3%). The difference between groups was significant regarding D-dimer (p= 0.0015) and ferritin (p= <0.0001), but not significant regarding CRP (p= 0.1233).” 3. Concern of the reviewer • Minor comment - I think the authors are saying that pregnant women are more vulnerable to infection and more likely to display a higher inflammatory response, exemplified by higher levels of D-dimer, etc. The way it is written makes it sound like the biomarkers are elevated first and then the infection occurs, which doesn’t really make sense from a biological perspective. The authors make a good point that it is important to compare these biomarkers among those in the ICU with and without COVID-19 to determine if there is something specific about being in the ICU alone that may influence D-Dimer, CRP, and ferritin. However, the point is a bit lost in this paragraph, so would be good to see it clarified. Our response:               Dear Reviewer #1, we appreciate your suggestion and the results paragraph was revised. Revised text:Page 5, lines 171-174, “perhaps pregnant women might be more vulnerable to SARS-CoV-2 infection, then presenting an easier chance to severe respiratory complications, which leads to intense inflammatory response exemplified by higher D-dimer, CRP and ferritin levels that may conduct to UCI cases.” 4. Concern of the reviewer • Results - Figure 3—all of the comparisons are significant, so it would be more useful to just include a footnote that makes that point instead of listing p-values for each of the comparisons. Our response:               Dear Reviewer #1, we appreciate your suggestion and the figure footnote was revised. Revised text:Page 8, line 257, “Significant correlations between biomarkers higher levels among pregnant with COVID-19 by trimesters.” 5. Concern of the reviewer • Tables - gestational age is broken into trimesters, not quarters (1st, 2nd, 3rd trimester). Our response:               Dear Reviewer #1, we appreciate your suggestion and the text in all tables containing quarters were adjusted.  6. Concern of the reviewer • Thank you for adding the overall totals with and without COVID-19 in Table 2. Please be sure to clarify in those last columns that they are the total with and without COVID-19. Our response:               Dear Reviewer #1, we appreciate your suggestion and the table was revised. Revised text:Page 7, lines 234-235,

With COVID-19

Without COVID-19

Total with and without COVID-19

P value

Ward

ICU

Ward

ICU

n

%

n

%

n

%

n

%

n

%

D dimer (ng/mL)

Normal (≤ 500)

9

12.5

1

3.8

22

29.3

12

40.0

44

21.6

Altered (501-1500)

34

47.2

12

46.1

21

28.0

11

36.6

78

38.4

0.0015a

Very altered (>1500)

29

40.2

13

50.0

32

42.6

7

23.3

81

39.9

CRP

Normal (≤ 8.0)

12

17.1

8

23.5

13

17.3

8

26.6

41

19.6

Altered (9-40)

19

27.1

5

14.7

30

40.0

6

20.0

60

28.7

0.1233b

Very altered (>40)

39

55.7

21

61.7

32

42.6

16

53.3

108

51.6

Ferritin

Normal (≤ 150)

32

50.7

4

17.3

60

80.0

22

73.3

118

61.7

Altered (151 – 450)

21

33.3

12

52.1

12

16.0

7

23.3

52

27.2

<0.0001a

Very altered (>450)

10

15.8

7

30.4

3

4.0

1

3.3

21

10.9

 7. Concern of the reviewer • Discussion – Please explicitly point out in the limitations that the study is underpowered and null findings for CRP should be interpreted with caution. That point is not clear in the revision..   Our response:               Dear Reviewer #1, we appreciate your suggestion and the limitation was modified.  Revised text:Page 10, lines 336-342, “As a limitation of this study, it should be highlighted that our sample size was relatively small, and it could underpowering our statistical associations importance and null findings for and COVID-19 severity among pregnant women in both groups, which it should be interpreted with caution. Also, due to our limitations we could not correlate elevated biomarkers are, specifically, caused by COVID-19, or if other factor or other infection could have been caused it.”

Reviewer 3 Report

The authors have answered to all questions and improved the manuscript according to the remarks.

Author Response

Dear Reviewer #3, we appreciate your text and suggestions to improve our paper and now if we fulfilled your considerations we would be thankful with the acceptance.